# Preoperative Serum Calcitonin and Its Correlation with Extent of Lymph Node Metastasis in Medullary Thyroid Carcinoma

**DOI:** 10.3390/cancers12102894

**Published:** 2020-10-09

**Authors:** Hyunju Park, Jun Park, Min Sun Choi, Jinyoung Kim, Hosu Kim, Jung Hee Shin, Jung-Han Kim, Jee Soo Kim, Sun Wook Kim, Jae Hoon Chung, Tae Hyuk Kim

**Affiliations:** 1Division of Endocrinology & Metabolism, Department of Medicine, Thyroid Center, Samsung Medical Center, Sungkyunkwan University School of Medicine, Seoul 06351, Korea; hj1006.park@samsung.com (H.P.); jun113.park@samsung.com (J.P.); minsun1.choi@samsung.com (M.S.C.); j513.kim@samsung.com (J.K.); sunwooksmc.kim@samsung.com (S.W.K.); thyroid@skku.edu (J.H.C.); 2Division of Endocrinology, Department of Medicine, Gyeongsang National University Changwon Hospital, Gyeongsang National University College of Medicine, Changwon 51472, Korea; narulake@naver.com; 3Department of Radiology, Samsung Medical Center, Sungkyunkwan University School of Medicine, Seoul 06351, Korea; helena35.shin@samsung.com; 4Division of Breast and Endocrine Surgery, Department of Surgery, Samsung Medical Center, Sungkyunkwan University School of Medicine, Seoul 06351, Korea; jinnee.kim@samsung.com (J.-H.K.); js0507.kim@samsung.com (J.S.K.)

**Keywords:** medullary thyroid carcinoma, calcitonin, lymph nodes, lymph node excision, clinical decision-making

## Abstract

**Simple Summary:**

Surgery is the only curative treatment for medullary thyroid carcinoma (MTC), but the initial surgical extent is still controversial. We examined whether the preoperative serum calcitonin level reflects the extent of lymph node metastasis (LNM), and therefore might be used to predict the optimal initial surgical extent for MTC. Furthermore, positive and negative likelihood ratios for preoperative serum calcitonin were calculated for calcitonin concentration categories, revealing that serum calcitonin levels can be of diagnostic value and might be applicable to surgical decision-making.

**Abstract:**

The optimal initial surgical extent for medullary thyroid carcinoma (MTC) remains controversial. Previous studies on serum calcitonin are limited to reporting the calcitonin threshold according to anatomical disease burden. Here, we evaluated whether preoperative calcitonin levels can be used to predict optimal surgical extent. We retrospectively reviewed the 170 patients with MTC at a tertiary Korean hospital from 1994 to 2019. We extracted data on preoperative calcitonin level, primary tumor size and the number and location of lymph node metastases (LNMs). To evaluate disease extent, we divided the patients into five groups: no LNM, central LNM, ipsilateral lateral LNM, contralateral lateral LNM, and distant metastasis. We calculated the positive and negative likelihood ratios (LRs) for multiple categories of preoperative calcitonin levels. Preoperative calcitonin level positively correlated with primary tumor size (rho = 0.744, *p <* 0.001) and LNM number (rho = 0.537, *p <* 0.001). Preoperative calcitonin thresholds of 20, 200, and 500 pg/mL were associated with the presence of ipsilateral lateral LNM, contralateral lateral LNM, and distant metastasis, respectively. The negative LRs were 0.1 at a preoperative calcitonin cut-off of 100 pg/mL in the central LNM, 0.18 at a cut-off of 300 pg/mL in the ipsilateral lateral LNM, and 0 at a cut-off of 300 pg/mL in the contralateral lateral LNM. The preoperative calcitonin level correlates with disease extent and has diagnostic value for predicting LNM extent. Our results suggest that the preoperative calcitonin level can be used to determine optimal initial surgical extent.

## 1. Introduction

Medullary thyroid carcinoma (MTC) accounts for 1–2% of all thyroid carcinomas in the United States, and 0.4–2.2% of all thyroid carcinomas in Korea [1,2,3]. The 10-year survival rate for the MTC is 75–85%, whereas the 10-year survival is about 90% for papillary thyroid carcinoma [4,5,6]. It occurs as both a sporadic and a familial disease, and in families, MTC can occur alone or in combination with multiple endocrine neoplasia type 2A (MEN-2A) or MEN-2B [2].

MTC arises from thyroid parafollicular cells (C cells), which secrete several hormones, including calcitonin. Calcitonin is a 32-amino acid monomeric peptide that is used as a highly sensitive and specific tumor marker for MTC [7,8]. The current American Thyroid Association (ATA) guidelines recommend measuring the serum calcitonin level when patients are diagnosed with histological MTC [9]. Additionally, serum calcitonin levels can be used in differential diagnosis, prognostic assessments, and evaluations of treatment response [10,11,12,13,14].

The standard initial treatment for MTC is surgery, and surgery is the only curative treatment option [9,15,16]. However, the extent of initial surgery for MTC is controversial because of the possibility of complications and its unknown benefits. The ATA guidelines recommend total thyroidectomy and the central compartment dissection of cervical lymph nodes, regardless of the presence of abnormal lymph nodes as determined by ultrasound [9]. By contrast, despite the general agreement that the therapeutic procedure for both sporadic and familial MTC is total thyroidectomy with central lymph node dissection, the National Comprehensive Center Network (NCCN) guidelines suggest that central neck dissection may not be necessary for MTCs ≤1 cm [15]. Based on a previous study reporting that ipsilateral and contralateral lateral lymph node metastasis (LNM) is associated with basal serum calcitonin levels exceeding 20 and 200 pg/mL, respectively [17], the current guidelines suggest that lateral lymph node dissection may be considered as a result of the serum calcitonin level and the results of ultrasound scans [9,16]. However, previous studies have reported threshold calcitonin levels depending only on the presence of LNM [12,17,18].

Here, we hypothesized that if the preoperative serum calcitonin level reflects the extent of LNM, the preoperative serum calcitonin level could also be a useful biomarker for tailoring the optimal initial surgical extent, particularly for lymph node dissection in the lateral neck compartment. Thus, we evaluated the correlation between preoperative serum calcitonin and the existence of LNM in MTC, and provide concrete stochastic information on the preoperative serum calcitonin level that could determine the optimal surgical extent.

## 2. Results

### 2.1. Baseline Characteristics and Factors Associated with Elevations in Preoperative Serum Calcitonin

The mean age of the 170 patients with MTC who had data on their preoperative serum calcitonin levels was 49.5 ± 14.5 years, and 111 patients (65.3%) were female. The median primary tumor size was 1.5 (0.7–2.5) cm, and the median preoperative serum calcitonin concentration was 401 (96.9–1398.5) pg/mL. The mean number of LNMs was 4.9 (8.6), and LNM was present in 148 (87.1%) patients. Among the patients with LNM, 72 (42.4%) patients, 62 patients (36.5%) and 14 patients (8.2%) had central LNM, ipsilateral lateral LNM, and contralateral lateral LNM, respectively. Eight patients (4.7%) had initial distant metastasis. Among the 170 patients in this study, 62 were classified as having a node stage above N1b. The group with a node stage below N1b had a higher proportion of females, a smaller primary tumor size, lower preoperative serum calcitonin levels and a lower number of LNMs compared to the group with a node stage above N1b (Table 1).

Because the preoperative serum calcitonin level was higher in the group with a node stage above N1b, we performed linear regression analyses to identify the factors that correlated with the preoperative serum calcitonin level. The multivariate linear regression analysis showed that log-calcitonin was significantly associated with primary tumor size (b = 0.33 and *p* < 0.001), node stage N1b (b = 0.47 and *p* < 0.001), and positive extrathyroidal extension (ETE) (b = 0.26 and *p* = 0.05) (Table 2).

### 2.2. Tumor Burden and Correlation with Preoperative Serum Calcitonin

Next, we assessed the relationships among the preoperative serum calcitonin levels, primary tumor size, and number of LNMs (Table 3). Because the preoperative serum calcitonin levels of the patients did not show a linear increase, we classified the serum calcitonin levels into eight categories, whereby category 1 was the lowest concentration range and category 8 was the highest. We found statistically significant relationships between the eight groups of increasing preoperative serum calcitonin levels and primary tumor size (rho = 0.744, *p* < 0.001) as well as the number of LNMs (rho = 0.537, *p* < 0.001).

Significant linear relationships were also observed between the preoperative serum calcitonin level and primary tumor size (R² = 0.459 and *p* < 0.001, Appendix A, supporting information) or the number of LNMs (R² = 0.201 and *p* < 0.001, Appendix A, supporting information). Subgroup analyses for sporadic MTC and hereditary MTC were also performed (Appendix A and S2b, supporting information). When we generated plot diagrams of the mean tumor size (Figure 1a) and number of LNMs (Figure 1b) against each calcitonin group, we noticed a sharp increase in tumor size when the calcitonin level exceeded 200.1 pg/mL (the lowest limit for the 4th group), and a sharp increase in the number of LNMs when the calcitonin level exceeded 1000.1 pg/mL (the lowest limit for the sixth group).

### 2.3. Tumor Extent According to the Preoperative Serum Calcitonin Level

We classified disease extent according to the region of LNM (central, ipsilateral lateral or contralateral lateral LNM) and distant metastasis. Preoperative serum calcitonin levels were significantly associated with tumor extent (*p* < 0.001). The post hoc analysis showed that ipsilateral lateral LNM, contralateral lateral LNM and distant metastasis were associated with higher calcitonin levels than no LNM or central LNM (Figure 2).

Patients with ipsilateral lateral LNM, contralateral lateral LNM and distant metastasis had preoperative serum calcitonin thresholds of 20.1, 200.1 and 500.1 pg/mL, respectively. However, one patient with distant metastasis had a preoperative serum calcitonin level less than 500.1 pg/mL (109 pg/mL). Central LNM was observed in all ranges of preoperative serum calcitonin levels (Table 4). Initial distant metastasis was observed in eight patients, and the most common site of distant metastasis was the lung. Detailed information on these patients is provided in Table 5.

### 2.4. LRs for the Preoperative Serum Calcitonin Level in Diagnosing the Extent of LNM

We calculated the likelihood ratios (LRs) to evaluate whether the calcitonin has diagnostic value as regards the disease extent. A positive LR greater than 10.0 or a negative LR less than 0.1 can be considered statistically adequate to confirm or exclude malignancy, respectively. Additionally, a positive LR between 5 and 10, and a negative LR between 0.1 and 0.2, are considered to have good diagnostic value. We calculated the LRs for multiple different levels of preoperative serum calcitonin (Table 6). We found that for patients with central LNM, the negative LR was 0.1 when the preoperative serum calcitonin level was less than 100 pg/mL, and that for patients with ipsilateral lateral LNM, the negative LR was 0.04, 0.12 or 0.18 when the preoperative serum calcitonin level was less than 100, 200 or 300 pg/mL, respectively. For patients with contralateral lateral LNM, the negative LR was 0 when the preoperative serum calcitonin level was less than 300 pg/mL (Table 6).

## 3. Discussion

Surgery is the only potentially curative treatment for MTC, but the initial surgical extent is controversial. Current guidelines suggest that serum calcitonin level could be considered when deciding the extent of surgery. This study found that preoperative serum calcitonin levels were closely associated with primary tumor size and the number of LNMs, and that increasing preoperative serum calcitonin was correlated with tumor extent. The thresholds of preoperative serum calcitonin according to disease extent were 20 pg/mL for ipsilateral lateral LNM, 200 pg/mL for contralateral lateral LNM, and 500 pg/mL for distant metastasis. Notably, the cut-off value for preoperative serum calcitonin with a negative LR ranging from 0.1 to 0.2 was less than 100 pg/mL in patients with central LNM and 300 pg/mL in patients with ipsilateral lateral LNM or contralateral lateral LNM. These findings indicate that the preoperative serum calcitonin level can serve as a practical biomarker to predict disease burden, namely, tumor size and number of LNMs, which is consistent with previous reports [11,13]. Furthermore, this study showed that the linear correlation between preoperative serum calcitonin category and tumor burden is significant regardless of whether the MTC was of the sporadic or hereditary type.

As biostatistical techniques have evolved over time, the use of statistical evaluation in biomarker exploration has become more important [19]. The strength of our study is that calcitonin also had diagnostic value for indicating the extent of LNM. The extent of lymph node dissection at the time of thyroidectomy is an ongoing discussion. Because the central node compartment is the site of primary lymphatic drainage from the thyroid, and LNM occurs during the early course of MTC [20,21], the guidelines of the ATA/European Thyroid Association (ETA), the British Thyroid Association (BTA), and the American Association of Endocrine Surgeons (AAES) recommend central node compartment dissection regardless of unequivocal evidence of cervical LNM [9,16,22]. However, the NCCN guidelines recommend total thyroidectomy with cervical dissection for primary tumors greater than 1.0 cm in diameter or bilateral thyroid disease [15]. Prophylactic lateral node dissection is more controversial. The preoperative serum calcitonin level and ultrasound findings can be helpful for determining the need for lateral node dissection [9,16]. Machens et al. reported preoperative serum calcitonin levels based on the pathologic extent of disease [17]. They suggested that preoperative serum calcitonin levels exceeding 20, 50 and 200 pg/mL were associated with ipsilateral central and lateral LNM, contralateral central LNM, and contralateral lateral LNM, respectively. However, they provided only a simple threshold of the calcitonin level that depended on the extent of metastasis. Based largely on the simple threshold results of a previous study, the ATA/ETA guidelines suggest that contralateral neck dissection should be considered when the serum calcitonin level is greater than 200 pg/mL. Additionally, the ATA/ETA guidelines recommend that dissection of the lateral neck compartment be determined based on serum calcitonin levels when patients have no evidence of neck metastasis on their ultrasound; however, they do not provide the exact cut-off calcitonin levels. The results from this study showed that preoperative serum calcitonin levels exceeding 20 and 200 pg/mL were associated with ipsilateral lateral LNM and contralateral lateral LNM, respectively, which is consistent with the findings of a previous study [17]. Furthermore, with our statistical analyses of patient data, we identified the preoperative serum calcitonin level that can be used as a diagnostic biomarker that reflects the disease extent. As a rule of thumb, a negative LR less than 0.1 indicates that a negative test is excellent at ruling out a diagnosis, and a negative LR less than 0.2 indicates that a negative test is good at ruling out a diagnosis [19,23,24].

Most of the MTC guidelines agree that the minimal standard initial therapeutic surgical extent is total thyroidectomy with central lymph node dissection. These surgical approaches may be reasonable, but long-term results are lacking. Notably, the relationship between the extent of cervical lymph node dissection and survival advantage has not been clearly defined [25]. Furthermore, central node dissection has the potential risks of hypoparathyroidism and recurrent laryngeal nerve injury [26]. This study revealed that the negative LR for central LNM was 0.1 when the preoperative serum calcitonin level was less than 100 pg/mL. Thus, consideration should be given to whether prophylactic central lymph node dissection should be performed in patients with a serum calcitonin level less than 100 pg/mL, and with no evidence of central node metastasis on the preoperative ultrasound. Likewise, the negative LR for ipsilateral lateral LNM was less than 0.2 when the preoperative serum calcitonin level was less than 300 pg/mL; therefore, prophylactic ipsilateral neck dissection in patients with no evidence of lateral node metastasis on their preoperative ultrasound also needs careful consideration. In particular, we found that the negative LR was 0 for contralateral lateral LNM when the preoperative serum calcitonin level was less than 300 pg/mL. Thus, when prophylactic contralateral neck dissection is being considered in patients with documented ipsilateral lateral LNM on their preoperative ultrasound, based on our study, a preoperative serum calcitonin level of 300 pg/mL can be used as a cut-off value. Here, we calculated negative LRs according to multiple categories of calcitonin levels for use in the clinic. We still believe that the purpose of treating patients with MTC is to achieve complete surgical resection in the early stages. Despite these goals, aggressive prophylactic lymph node dissection should be considered carefully only when the need is great. Thus, experienced practitioners may use an alternative cut-off level of preoperative serum calcitonin, depending on the clinical situation.

According to the ATA, preoperative ultrasound evaluation is the most important preoperative imaging study [9]. However, the accuracy of ultrasound is highly dependent on experience and expertise [27,28]. False-negative results from preoperative ultrasounds may lead to inadequate surgical management that increases the risk of recurrence and a poor prognosis. Thus, if there is any doubt that preoperative ultrasound was performed properly, we propose that the negative LR according to the preoperative serum calcitonin level can be used to predict the extent of LNM so as to optimize the initial surgery.

This study has some limitations. One of the critical limitations of this study is that this was a retrospective design conducted in a single tertiary referral center, and accordingly selection bias might have occurred. Nonetheless, the study population included a relatively large number of patients, despite the low incidence rate of MTC. The distant metastasis group, however, is composed of a relatively small number of patients. Thus, further prospective multicenter studies are needed to validate the current results. Second, guidelines evolved during the study period. Most of the enrolled patients underwent total thyroidectomy with initial central lymph node dissection, which is consistent with current guidelines. However, 11 patients (6.5%) out of the enrolled patients did not undergo initial central lymph node dissection. To overcome this problem, pre- and postoperative thyroid ultrasound, computer tomography (CT), postoperative serum calcitonin levels, and the clinical course for each patient were reviewed to rule out the possibility of micrometastases (supplementary Appendix A). All these patients had no evidence of lymph node metastasis in their preoperative images. Of 11 patients, 1 patient was transferred out to another hospital after surgery, but this patient only had micro-MTC. Other patients had postoperative image studies done, and there was still no evidence of lymph node metastasis after thyroidectomy. Furthermore, eight patients achieved biochemical cure during the follow-up. Among the other two patients who were not biochemically cured, one of them had recurrence at the thyroid operative bed after 9 years of follow-up, and another had no evidence of disease during 8 years of follow-up. Overall, the possibility of missed micrometastases at the first surgery seems to be low. Third, in patients with advanced MTC, carcinoembryonic antigen (CEA) is considered as a useful biomarker for evaluation of disease progression, and for monitoring patients following thyroidectomy [9]. We tried to evaluate the CEA as a diagnostic biomarker biostatistically, but because of the retrospective nature of this study, we could not get enough information about the preoperative serum CEA level.

## 4. Materials and Methods

### 4.1. Study Population

In total, 246 patients who were diagnosed with MTC at the Samsung Medical Center between 1994 and 2019 were included in this study. Of these patients, 170 had preoperative serum calcitonin measurements available for analysis. Their clinical and pathology reports were obtained and retrospectively reviewed. This study was approved by the institutional review board of Samsung Medical Center (IRB No. 2020-07-007), and patient consent was waived in instances by the committee, because of the retrospective chart review study design and the use of only deidentified clinicopathological information.

### 4.2. Analysis of Metastasis and Tumor Extent

From the pathology reports, we obtained information on the primary tumor diameter (cm), location and total number of LNMs, ETE, tumor extent and distant metastasis. The extent of LNM was categorized according to location: central LNM, ipsilateral lateral LNM or contralateral lateral LNM. Central LNM included both ipsilateral and contralateral LNM. Tumor extent was classified into five groups according to the presence of LNM and distant metastasis: (1) no LNM, (2) the presence of central LNM without lateral LNM or distant metastasis, (3) the presence of ipsilateral lateral LNM without contralateral lateral LNM or distant metastasis (regardless of the central LNM status), (4) the presence of contralateral lateral LNM without distant metastasis (regardless of the central LNM status), and (5) the presence of distant metastasis (regardless of LNM). Initial distant metastasis was defined as distant metastases that were detected within 6 months after the initial surgery. Distant metastatic lesions were detected by chest and/or abdominopelvic CT, magnetic resonance imaging (MRI), whole-body bone scintigraphy, and 19-fluorodeoxyglucose positron emission tomography (FDG-PET), and/or were pathologically confirmed. For anatomical staging, the 8th edition of the American Joint Cancer Committee (AJCC) system was used.

### 4.3. Measurement of Serum Calcitonin

The serum calcitonin concentration was measured with a two-site immunoradiometric assay using a commercial kit (MEDGENIX CT-U.S.-IRMA kit, BioSource Europe S.A., Belgium). The minimum detectable limit was 0.9 pg/mL. All samples were measured in duplicate, and the intraassay and interassay coefficients of variation were 2.4–3.4% and 3.6–5.4%, respectively.

Preoperative serum calcitonin was divided into eight categories considering the number of patients and previous reported calcitonin threshold according to disease extent [17]: category 1 (1–20 pg/mL), category 2 (20.1–100 pg/mL), category 3 (100.1–200 pg/mL), category 4 (200.1–500 pg/mL), category 5 (500.1–1000 pg/mL), category 6 (1000.1–2000 pg/mL), category 7 (2000.1–7000 pg/mL), and category 8 (more than 7000 pg/mL).

### 4.4. Statistical Analysis

Descriptive statistics were analyzed using Student’s t-test or the Kruskal–Wallis test as appropriate, and are presented as means ± standard deviations (SDs) or medians and interquartile ranges (IQRs). One-way ANOVA or the Mann–Whitney U test with Bonferroni’s method were performed for the post hoc analysis of descriptive statistics. Categorical variables were analyzed using the chi-square test or Fisher’s exact test and are presented as absolute numbers (percentages). Spearman correlation analysis was used to investigate the strength of an association between two continuous variables, and is represented by the correlation coefficient, rho. Linear regression analysis was performed to assess variables that could be associated with the serum calcitonin elevation in univariable and multivariable analysis. To assess continuous serum calcitonin levels, transformed calcitonin with common log (log-calcitonin) was used due to non-normality in the linear regression analysis. Positive and negative LRs were calculated using an established method [19,23]. A *p* value < 0.05 was considered to indicate statistical significance. Statistical analysis was performed using SPSS version 25.0 for Windows (IBM, Chicago, IL, USA).

## 5. Conclusions

The preoperative serum calcitonin level correlated with disease extent and showed excellent diagnostic value for predicting the concrete extent of LNM. Our results suggest that preoperative serum calcitonin levels can be used to determine the optimal initial surgical extent to reduce the number of reoperations while minimizing surgical complications.

## Figures and Tables

**Figure 1 cancers-12-02894-f001:**
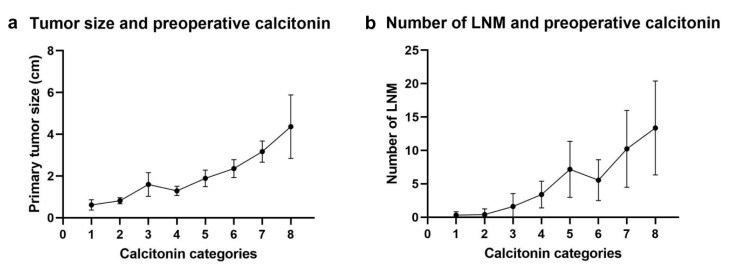
Correlation between tumor extent and preoperative serum calcitonin levels. (**a**) Size of the primary tumor. (**b**) Number of LNMs (Calcitonin categories: category 1 (1–20 pg/mL), category 2 (20.1–100 pg/mL), category 3 (200.1–500 pg/mL), category 4 (200.1–500 pg/mL), category 5 (500.1–1000 pg/mL), category 6 (1000.1–2000 pg/mL), category 7 (2000.1–7000 pg/mL) and category 8 (more than 7000 pg/mL). The point represents the mean of the each category, and the error bars represent 95% confidential interval).

**Figure 2 cancers-12-02894-f002:**
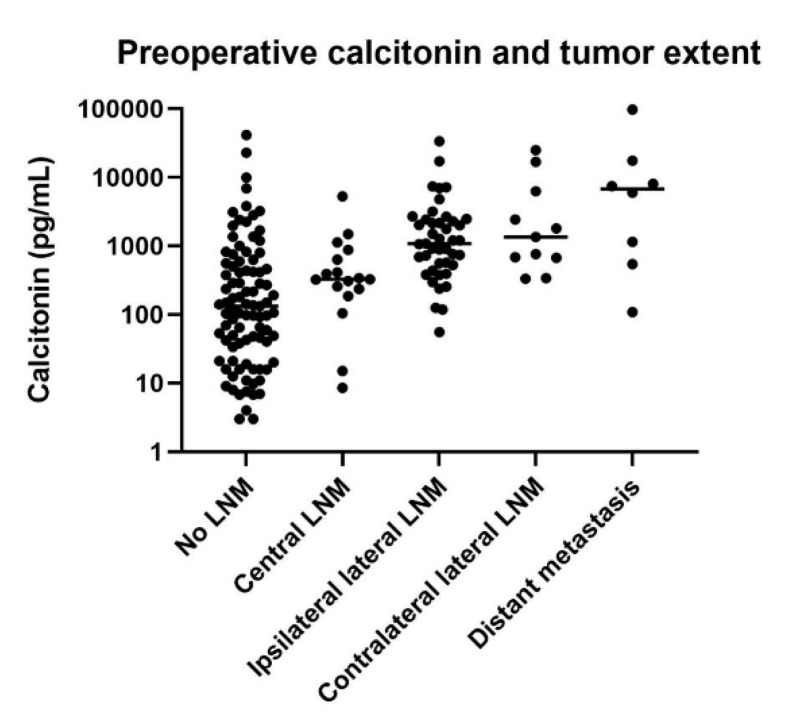
Correlation between tumor extent and preoperative serum calcitonin levels. The dots represent serum calcitonin level in each patient, and the horizontal lines represent the mean of the group.

**Table 1 cancers-12-02894-t001:** Baseline clinicopathologic characteristics of the patients.

Characteristic	Number of Patients (*n =* 170)	Below N1b (*n =* 108)	Above N1b (*n =* 62)	*p* Value
Age (mean ± SD)	49.5 ± 14.5	50.5 ± 14.0	47.7 ±15.4	0.235
Sex (%)				
Female	111 (65.3)	78 (72.2)	33 (53.2)	0.012
Male	59 (34.7)	30 (27.8)	29 (46.8)	
Tumor type (%)				
Sporadic	139 (81.8)	85 (78.7)	54 (87.1)	0.173
Hereditary (MEN-2A)	31 (18.2)	23 (21.3)	9 (12.9)	
Primary tumor size (median, IQR)	1.5 (0.7–2.5)	1.2 (0.6–1.9)	2.0 (1.3–3.5)	<0.001
Preoperative serum calcitonin (pg/mL, median, IQR)	401.0 (96.93–1398.5)	162.5 (40.8–587.0)	1201.0 (554.3–2827.0)	<0.001
Extrathyroidal extension (%)				
Negative	125 (73.5)	98 (90.7)	27 (43.5)	<0.001
Positive	45 (26.5)	10 (9.3)	35 (56.5)	
Number of LNMs (mean ± SD)	4.9 (8.6)	0.3 (1.0)	12.8 (10.1)	<0.001
Extent of LNM (%)				
Central LNM	72/170 (42.4)	17/108 (15.7)	55/62 (88.7)	<0.001
Ipsilateral lateral LNM	62/170 (36.5)	0	62/62 (100)	<0.001
Contralateral lateral LNM	14/170 (8.2)	0	14/62 (22.6)	<0.001
Distant metastasis (%)	8 /170(4.7)	0	8/62 (12.9)	<0.001
AJCC 8th stage (%)				
1	68 (40.0)	68 (63.0)	0 (0)	<0.001
2	23 (13.5)	23 (21.3)	0 (0)	
3	17 (10.0)	17 (15.7)	0 (0)	
4a	54 (31.8)	0 (0)	53 (85.5)	
4b	0 (0)	0 (0)	1 (1.6)	
4c	8 (4.7)	0	8 (12.9)	

N1b = Presence of unilateral, bilateral, or contralateral lateral lymph node metastasis; SD = standard deviation; IQR = interquartile range; MEN-2A = multiple endocrine neoplasia type 2A; LNM = lymph node metastasis; AJCC = American Joint Committee on Cancer.

**Table 2 cancers-12-02894-t002:** Linear regression analysis of preoperative serum calcitonin (pg/mL, log-transformed) and the indicated factors.

Characteristic	Unadjusted		Adjusted	
	b ± SE	*p* Value	b ± SE	*p* Value
Sex (female)	0.137 ± 0.103	0.188		
Age at diagnosis (years)	0.006 ± 0.004	0.083		
Primary tumor size	0.328 ± 0.035	<0.001	0.327 ± 0.035	<0.001
Nodal metastasis above N1b	0.529 ± 0.121	<0.001	0.472 ± 0.118	<0.001
Positive extrathyroidal extension	0.265 ± 0.131	0.045	0.257 ± 0.131	0.050
Positive resection margin	−0.057 ± 0.457	0.901		
Tumor type (sporadic/hereditary)	0.239 ± 0.134	0.076		

SE = standard error.

**Table 3 cancers-12-02894-t003:** Tumor extent and cancer prognosis according to preoperative serum calcitonin levels.

Calcitonin Category	Preoperative Serum Calcitonin (pg/mL)	Number of Patients	Primary Tumor Size		Number of LNMs	
			Mean	95% CI	Mean	95% CI
1	1–20	22	0.61	0.36–0.86	0.32	0–0.82
2	20.1–100	22	0.81	0.67–0.96	0.41	0–1.26
3	100.1–200	18	1.59	1.03–2.16	1.61	0–3.56
4	200.1–500	30	1.29	1.07–1.51	3.4	1.40–5.4
5	500.1–1000	24	1.89	1.50–2.29	7.17	2.96–11.35
6	1000.1–2000	18	2.36	1.93–2.78	5.56	2.47–8.64
7	2000.1–7000	22	3.17	2.66–3.68	10.23	4.46–16.0
8	More than 7000	14	4.09	2.57–5.60	13.36	6.36–20.38
Total		170	1.87	1.64–2.10	4.89	3.59–6.19

LNM = lymph node metastasis; CI = confidence interval.

**Table 4 cancers-12-02894-t004:** Lymph node metastasis and distant metastasis by preoperative serum calcitonin levels.

Calcitonin Category	Preoperative Serum Calcitonin (pg/mL)	Number of Patients	Central LNM		Ipsilateral Lateral LNM		Contralateral Lateral LNM		Distant Metastasis	
			*n*	%	*n*	%	*n*	%	*n*	%
1	1–20	22	2	9.1	0	0	0	0	0	0
2	20.1–100	22	1	4.5	1	4.5	0	0	0	0
3	100.1–200	18	5	27.8	3	16.7	0	0	1	5.6
4	200.1–500	30	17	56.7	9	30.0	2	6.7	0	0
5	500.1–1000	24	15	62.5	14	58.3	3	12.5	1	4.2
6	1000.1–2000	18	12	66.7	10	55.6	2	11.1	1	5.6
7	2000.1–7000	22	12	54.5	14	63.6	3	13.6	1	4.5
8	More than 7000	14	8	57.1	11	78.6	4	28.6	4	28.6
Total		170	72	42.4	62	36.5	14	8.2	8	4.7

LNM = lymph node metastasis.

**Table 5 cancers-12-02894-t005:** Detailed information of patients with distant metastases.

No.	Age at Dx	Sex	Sporadic/Hereditary	Preoperative Serum Calcitonin (pg/mL)	Primary Tumor Size (cm)	Number of LNMs	Central LNM	Ipsilateral Lateral LNM	Contralateral Lateral LNM	N Stage	Site of the Initial Distant Metastasis
1	21	M	Sporadic	109	4.5	14	Positive	Positive	Negative	N1b	Lung
2 ^†^	47	M	Sporadic	543	1.5	26	Positive	Positive	Negative	N1b	Lung
3	39	F	Sporadic	1150	0.9	11	Positive	Positive	Positive	N1b	Lung
4	43	M	Sporadic	6012	2.8	38	Positive	Positive	Negative	N1b	Lung
5	63	M	Sporadic	7488	7.0	5	Negative	Positive	Negative	N1b	Mediastinal soft tissue
6	26	F	Sporadic	8060	4.1	26	Positive	Positive	Positive	N1b	Liver
7	55	M	Sporadic	17,450	2.6	14	Positive	Positive	Negative	N1b	Lung, liver
8	24	F	Sporadic	97,500	8.0	10	Positive	Positive	Positive	N1b	Lung, bone

No. = patient number; Dx = diagnosis; LNM = lymph node metastasis. ^†^ This patient did not harbor a germline RET gene alteration in his blood sample analysis, which is routinely performed in this hospital. Upon progression, his tumor specimen was subjected to a screening test for somatic RET gene alterations, and he has been enrolled in a clinical trial for RET-positive solid tumors since November 2019.

**Table 6 cancers-12-02894-t006:** Likelihood ratios for preoperative serum calcitonin levels in diagnosing lymph node metastasis and distant metastasis.

Calcitonin Cut-Off (pg/mL)	N with Central LNM	N without Central LNM	Positive LR	Negative LR	N with IpsilateralLateral LNM	N without Ipsilateral Lateral LNM	Positive LR	Negative LR
20	2	20	1.22 (1.10–1.36)	**0.14 (0.03–0.56)**	0	22	1.26 (1.14–1.38)	0
100	3	41	1.65 (1.38–1.96)	**0.10 (0.03–0.31)**	1	43	1.64 (1.40–1.91)	**0.04 (0.01–0.29)**
200	8	54	1.98 (1.57–2.50)	0.20 (0.10–0.40)	4	58	2.02 (1.63–2.50)	**0.12 (0.05–0.32)**
300	13	61	2.17 (1.65–2.86)	0.29 (0.17–0.49)	7	67	2.33 (1.81–3.02)	**0.18 (0.09–0.37)**
400	23	62	1.86 (1.37–2.51)	0.50 (0.35–0.73)	12	73	2.53 (1.89–3.40)	0.27 (0.16–0.45)
500	25	67	2.07 (1.47–2.89)	0.51 (0.36–0.72)	13	79	2.94 (2.10–4.12)	0.29 (0.17–0.47)
1000	40	76	1.98 (1.26–3.10)	0.72 (0.57–0.90)	27	89	3.21 (2.02–5.10)	0.53 (0.39–0.71)
2000	52	82	1.71 (0.95–3.05)	0.86 (0.73–1.02)	37	97	3.95 (2.09–7.48)	0.67 (0.54–0.82)
3000	59	88	1.78 (0.82–3.81)	0.91 (0.80–1.04)	47	100	3.27 (1.47–7.26)	0.82 (0.70–0.95)
5000	61	91	2.16 (0.87–5.25)	0.91 (0.82–1.02)	49	103	4.50 (1.70–12.10)	0.83 (0.72–0.95)
7000	64	92	1.82 (0.66–5.00)	0.95 (0.86–1.04)	51	105	**6.32 (1.85–22.02)**	0.85 (0.75–0.95)
10,000	68	94	1.37 (0.35–5.26)	0.98 (0.92–1.06)	56	106	**5.11 (1.09–25.11)**	0.92 (0.85–1.00)
Total	72	98			62	108		
**Calcitonin** **cut-off** **(pg/mL)**	**N with Contralateral Lateral LNM**	**N without** **Contralateral Lateral LNM**	**Positive LR**	**Negative LR**	**N with** **Distant** **Metastasis**	**N without** **Distant** **Metastasis**	**Positive LR**	**Negative LR**
20	0	22	1.16 (1.09–1.24)	**0**	0	22	1.16 (1.09–1.23)	**0**
100	0	44	1.40 (1.26–1.54)	**0**	0	44	1.37 (1.25–1.51)	**0**
200	0	62	1.66 (1.46–1.89)	**0**	1	61	1.40 (1.05–1.87)	0.33 (0.05–2.10)
300	0	74	1.90 (1.64–2.21)	**0**	1	73	1.59 (1.18–2.14)	0.28 (0.04–1.75)
400	2	83	1.83 (1.40–2.40)	0.27 (0.07–0.98)	1	84	1.82 (1.34–2.47)	0.24 (0.04–1.52)
500	2	90	2.03 (1.53–2.69)	0.25 (0.07–0.90)	1	91	2.00 (1.46–2.74)	0.22 (0.04–1.40)
1000	5	111	2.23 (1.40–3.54)	0.50 (0.25–1.02)	2	114	2.53 (1.59–4.03)	0.36 (0.11–1.19)
2000	7	127	2.69 (1.45–4.99)	0.61 (0.36–1.04)	3	131	3.27 (1.75–6.09)	0.46 (0.19–1.14)
3000	8	139	3.94 (1.85–8.35)	0.64 (0.41–1.01)	3	144	**5.63 (2.82–11.23)**	0.42 (0.17–1.03)
5000	8	144	**5.57 (2.47–12.57)**	0.62 (0.39–0.98)	3	149	**7.81 (3.69–16.46)**	0.41 (0.17–1.00)
7000	10	146	**4.47 (1.60–12.39)**	0.76 (0.55–1.07)	4	152	**8.07 (3.24–20.26)**	0.53 (0.27–1.07)
10,000	11	151	**6.69 (1.78–25.10)**	0.81 (0.62–1.07)	6	156	**6.76 (1.61–28.33)**	0.78 (0.52–1.16)
Total	14	156			8	162		

N = number of patients; LNM = lymph node metastasis; LR = likelihood ratio; Negative LR below 0.2 and Positive LR above 5 were marked in bolds number.

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
