# Peer review of "Preoperative Serum Calcitonin and Its Correlation with Extent of Lymph Node Metastasis in Medullary Thyroid Carcinoma"

_cancers, 2020, doi:10.3390/cancers12102894_

Round 1

Reviewer 1 Report

The authors propose a possible correlation between pre-surgical serum calcitonin levels and medullary thyroid cancer extent.

The topic is interesting, because the diagnostic and prognostic capacity of serum calcitonin at diagnosis is certainly still controversial. Or at least the cutoffs have yet to be defined uniquely.

  However, the present study has important limitations that prevent the results from supporting the conclusions proposed by the authors.   Major comments: a) First of all, the authors clearly state that not all subjects were treated with lateral compartment dissection and some did not even undergo central compartment dissection. This is correct and understandable from a clinical point of view. However, to study a correlation between calcitonin and lymph node metastases, it is essential to have an exhaustive characterization of the lymph nodes of all subjects. How can the authors rule out the presence of micrometastases in subjects who did not perform lymphnode dissection? They claim to have considered pre-surgery ultrasound, but it is known that metastases of the central compartment are difficult to detect with the thyroid in place. b) Figure 2: it is evident that the group "NO LMN" has a very wide range of calcitonine levels. Maybe  because it contains subjects who have not been treated with dissection but who had metastases? c) I suggest to calculate thresholds with AUC d) any information about adverse events related to surgery in these patients?
any details about node metastases (size)?   Minor comments: a) line 27: "is still" instead of "was" b) line 59: "MTC arises from thyroid follicular cells". The error of this sentence is so gross that it demonstrates serious superficiality in the writing. Or the authors ignore that MTC arises from PARAfollicular cells and not from follicular cells? c) line 120: "was"? d) line 248: also endocrinologists perform ultrasounds 

Reviewer 2 Report

In this manuscript, the authors evaluated whether preoperative calcitonin levels could be used to predict optimal surgical extent in 170 patients with Medullary Thyroid Carcinoma. The paper is well written and it is easily to understand. Additionally, the results are supported by statistical analysis and clinical measures. However, there are some major corrections that authors should modify to publish this paper in Cancers journal.

  1. In the abstract no abbreviation of MTC in the fourth line was included. Please add it.
  2. In the results section, the upper range of median preoperative serum calcitonin concentration has a mistake. Please correct.
  3. Also in the results section, "Among the patients with LNM, 72 (42.4%) patients, 62 patients (36.5%), and 14 patients 90 (8.2%)". Please put the first percentage in the correct order.
  4. In figure 2, the data about distance metastasis and calcitonin levels is very scarce and disseminated. Indeed, there are more values below the median that over that value. Maybe more patients should be included in this group to confirm those results.
  5. There must be some mistakes in Table 4. I do not understant n and percentages in all the groups. In addition, the group of distant metastases has lower levels of calcitonin than the rest of groups. For example, only 28% had more than 7000 pg/mL; whereas central and ipsilateral LNM had 57 and 78% respectivelly. The same happens from 2000-7000. Metastasis group only accounts 4.5%; whereas the rest of groups had 54%, 63% and 13%. Maybe, metastasis and calcitonin levels are not associated as the authors state. Some explanations should be provided by the authors.

Reviewer 3 Report

The paper was aimed to assess the preoperative serum calcitonin (Ct) concentration as a predictor of lymph node metastases (LNM) in medullary thyroid carcinoma (MTC) advancement.

The authors demonstrated that the preoperative serum Ct level reflects the extent of LNM and presence of distant metastasis, and could also be a useful biomarker for tailoring the optimal initial surgical extent, particularly for lymph node dissection in the lateral neck compartment.

The presented results bring nothing new to the current knowledge. This is very well known that Ct concentration is a very sensitive marker of tumour mass. Many researchers extensively analyse the correlation between the serum Ct level and the frequency and pattern of lymph node metastases and many scientific societies published this information in their guidelines.

This research has serious flaws, especially the small number of patients (170) and its retrospective character - authors reviewed the patients with MTC who were treated from 1994 to 2019.

Additionally, this is rather unlikely that patients with Ct concentration above than 400 pg/mL were free of LNM or distant metastasis (Figure 2). In my opinion it is possible that they had not been diagnosed correctly.

Reviewer 4 Report

The authors evaluated preoperative Serum Calcitonin and Its Correlation  with Extent of Lymph Node Metastasis in Medullary Thyroid Carcinoma.

I kindly want to have attention only 2 points:

  • In table 3, Tumour extent and cancer prognosis according to preoperative serum calcitonin levels, the authors classified the serum calcitonin levels are colleced in 8 categories. How the authors decided to these 8 intervals for each category?
  • Do the authors use CEA levels in the routine evaluation of these patients? Any avaliable data about it? Or any comment in the discussion part? (Wells SA Jr, Asa SL, Dralle H, Elisei R, (Evans DB, Gagel RF, Lee N, Machens A, Moley JF, American Thyroid Association Guidelines Task Force on Medullary Thyroid Carcinoma et al (2015) Revised American Thyroid Association guidelines for the management of medullary thyroid carcinoma. Thyroid 25:567–610 AND Machens A, Dralle H (2010) Biomarker-based risk stratification for previously untreated medullary thyroid cancer. J Clin Endocrinol Metab 95(6):2655–2663. https://doi.org/10.1210/jc.2009-2368)

Reviewer 5 Report

Hyunju et al study the role of calcitonin in determining the extent of LN dissection in MTC. Overall, this is a well-conducted and scientifically sound study. The data are well presented and well described. This work is the result of analysis from a patient population with a relatively rare tumor type at one institution, making the findings potentially non-generalizable. However, the authors include these caveats in their discussion and provide ample statistical proof to substantiate the potential for these findings to be generalizable to other populations. Additionally, the biological relationships behind these findings may be applicable to other types of tumors.

The strengths of this paper include the significance of the findings and potential applicability to clinical practice. All of the presented data support the central findings of the study, namely that pre-operative calcitonin levels predict tumor size and tumor spread. This means that pre-operative calcitonin levels can be used to guide surgical treatment decision in patients with medullary thyroid carcinoma. I look forward to seeing these findings re-tested in larger and more diverse patient populations.

The inclusion of the un-binned correlation data between pre-operative calcitonin and both tumor size and tumor spread as a supplementary figure adds considerably to the manuscript. However, I agree with the inclusion of this figure as supplementary, as the binned data is more appropriate for the main body of the paper. 

I commend the authors on their valuable analysis and manuscript. I would recommend publication of this manuscript after minor revisions, primarily focusing on the few typos found within the paper. Overall, the rigor of this analysis and these findings merit publication in Cancers.

There are a few minor typos.

Round 2

Reviewer 1 Report

The authors have provided some satisfactory answers to my requests, improving the manuscript. However, the methodological limits intrinsic to the study and not surmountable (for example retrospective nature with lack of some data) do not, in my opinion, make it worthy of publication in a journal like Cancers

Reviewer 2 Report

The authors have answered all my questions.
